# Interpretable Self-Aware Neural Networks for Robust Trajectory Prediction

**Masha Itkina and Mykel J. Kochenderfer**
Department of Aeronautics and Astronautics, Stanford University
{mitkina,mykel}@stanford.edu

**Abstract:** Although neural networks have seen tremendous success as predictive models in a variety of domains, they can be overly confident in their predictions on out-of-distribution (OOD) data. To be viable for safety-critical applications, like autonomous vehicles, neural networks must accurately estimate their epistemic or model uncertainty, achieving a level of system *self-awareness*. Techniques for epistemic uncertainty quantification often require OOD data during training or multiple neural network forward passes during inference. These approaches may not be suitable for real-time performance on high-dimensional inputs. Furthermore, existing methods lack interpretability of the estimated uncertainty, which limits their usefulness both to engineers for further system development and to downstream modules in the autonomy stack. We propose the use of evidential deep learning to estimate the epistemic uncertainty over a low-dimensional, interpretable latent space in a trajectory prediction setting. We introduce an interpretable paradigm for trajectory prediction that distributes the uncertainty among the semantic concepts: past agent behavior, road structure, and social context. We validate our approach on real-world autonomous driving data, demonstrating superior performance over state-of-the-art baselines. Our code is available at: https://github.com/sisl/InterpretableSelfAwarePrediction.

**Keywords:** Autonomous vehicles, trajectory prediction, distribution shift

## 1 Introduction

Deep learning techniques have had success across a multitude of domains, including human trajectory prediction in the context of autonomous vehicles (AVs) [1, 2]. However, a key challenge in deploying these systems is their lack of *self-awareness* about the quality of their predictions. Deep learning models often overestimate their confidence in unfamiliar situations [3–9]. For robots deployed in human environments, this over-confidence could result in dangerous maneuvers and safety concerns. Flagging unfamiliar situations encountered in the real world could be helpful to downstream autonomy stack components, such as path planners to execute a fail-safe maneuver, and to engineers for further system development.

Real-world systems are subject to *aleatoric* and *epistemic* uncertainty. The former is irreducible data uncertainty, which, for trajectory prediction, could be represented as a distribution over future trajectories given past observations (e.g., turning or continuing straight given a straight past trajectory). Aleatoric uncertainty can be modelled explicitly during training of learning-based systems, for example, using variational approaches [10–16]. Epistemic uncertainty reflects *what the model does not know*, which can arise due to the limited ability of a model to represent data and data distribution shift. Neural networks often fail to correctly calibrate this uncertainty, resulting in unreliable predictions for out-of-distribution (OOD) inputs [3–9]. In this paper, we focus on epistemic uncertainty quantification for trajectory prediction.

There has been growing interest in estimating epistemic uncertainty for deep learning models [17]. Most existing methods consider small benchmark datasets or require OOD data for training [18–23]. A few recent papers (e.g., [24–31]) have began exploring epistemic uncertainty estimation for learning-based robot perception and prediction. Identifying OOD inputs for tasks like trajectory prediction is difficult due to the high dimensionality of the data. Since it is impossible to foresee

6th Conference on Robot Learning (CoRL 2022), Auckland, New Zealand.

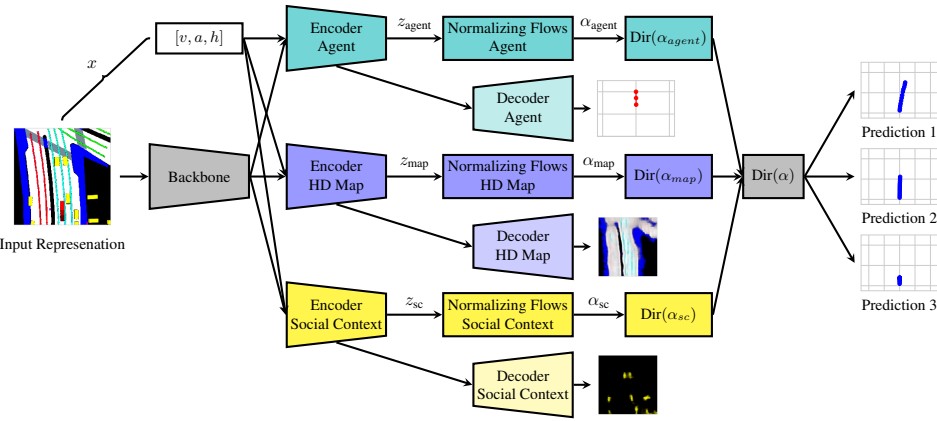

Figure 1: Interpretable self-aware prediction (ISAP) system overview. ISAP learns the uncertainty distributed over semantically interpretable latent concepts: the agent's past behavior, map, and social context. The network outputs parameters $\alpha_{agent}$, $\alpha_{map}$, and $\alpha_{sc}$ for the corresponding Dirichlet distributions, learned using normalizing flows, which can then be combined to form the output Dirichlet distribution, $\text{Dir}(\alpha)$. The aleatoric uncertainty for the trajectory prediction task is modeled by the expected value of the Dirichlet parameters $\alpha$ and can be used to make trajectory predictions (three most likely predictions are shown), while their sum $\alpha_0$ is an indicator of epistemic uncertainty. A high $\alpha_0$ indicates evidence for an input, and corresponds to low epistemic uncertainty.

all possible scenarios encountered on the road, an OOD dataset cannot be manually curated for training. Moreover, successful epistemic uncertainty estimation techniques, such as ensembles [32] and Monte Carlo (MC) dropout [18], require multiple forward passes during inference, potentially hindering real-time performance. Finally, most methods output a single value for epistemic uncertainty. In a cluttered, dynamic setting (e.g., urban driving), where there are multiple possible sources of uncertainty, this value could be challenging to interpret. For example, it may be unclear whether high epistemic uncertainty stems from an unfamiliar trajectory maneuver, new intersection type, or strange behavior of surrounding agents due to an external disturbance (e.g., road construction).

Modeling epistemic uncertainty for trajectory prediction has three key challenges: (1) lack of explicit OOD data during training, (2) efficiency requirements during inference, and (3) interpretability of the computed uncertainty. This paper addresses these challenges for trajectory prediction through an evidential deep learning approach that estimates the epistemic uncertainty in one-shot with only a single forward pass through the model and requires no OOD data for training. We allocate this uncertainty among interpretable, low-dimensional latent variables. *Evidential deep learning* estimates parameters for a second-order distribution (e.g., a Dirichlet distribution) to capture the epistemic uncertainty [19, 20, 33]. To avoid OOD data for training, normalizing flows may be used to constrain the learned Dirichlet distribution parameters by enforcing the densities for each class to integrate to the number of training samples in that class, as in the Posterior Network (PostNet) architecture [34].

In trajectory prediction, human behavior is often modeled using discrete modes that can represent high-level maneuvers like accelerating, braking, and turning [12, 35–39]. We apply ideas from PostNet [34] for modeling epistemic uncertainty to trajectory prediction architectures with discrete modes. To address interpretability in the learned epistemic uncertainty, we propose the following semantic insight. High epistemic uncertainty for trajectory prediction may originate from unfamiliar input behaviors, road structures, or social contexts. We distribute the learned epistemic uncertainty over these categories encoded in a low-dimensional latent space, thus encouraging tractability. Hence, we introduce a trajectory prediction paradigm that is self-aware of its prediction confidence and able to provide rich, interpretable information to downstream autonomy stack components and to engineers for development. We call this paradigm *Interpretable Self-Aware Prediction (ISAP)*.

Our key contributions are: 1) We propose a novel application of evidential deep learning to the task of epistemic uncertainty quantification for trajectory prediction. 2) We introduce interpretability into the uncertainty estimate by distributing it over interpretable, low-dimensional latent variables. The epistemic uncertainty source is decomposed as coming from unfamiliar agent behavior, road configuration, or social context. 3) We demonstrate superior uncertainty estimation performance of our ISAP framework over state-of-the-art (SOTA) approaches on the real-world NuScenes dataset [40].

## 2 Related Work

**Interpretable Trajectory Prediction.** One way to encourage interpretability in trajectory prediction architectures is through discrete modes [12, 36–39, 41]. For example, Chai et al. [37], Mangalam et al. [42], and Hu et al. [43] induce interpretability by learning distributions over discrete intention goals. Kothari et al. [41] learn a probability distribution over possibilities in an interpretable discrete choice model. Due to the ubiquity of discrete modes in trajectory prediction literature, we develop an epistemic uncertainty quantification approach assuming the presence of discrete modes in the architecture. Another common method for facilitating interpretability in a latent space is through an encoder-decoder structure. Neumeier et al. [44] use a decoder with expert knowledge to produce an interpretable latent space in a trajectory prediction model. Inspired by this idea, we enforce the epistemic uncertainty to be learned over three separate latent encodings that correspond to observed agent behavior, road configuration, and social context. This interpretable structure is achieved by using decoder components that are learned through self-supervised signals.

**Epistemic Uncertainty for Learning-Based Perception and Prediction.** Gawlikowski et al. [17] survey modern uncertainty quantification methods. A common approach to compute epistemic uncertainty in learning-based autonomous systems is to use MC dropout [18] due to its simple implementation and because it does not require OOD data for training. This approach is used in tasks like trajectory prediction [45, 46], pedestrian bounding box prediction [47], semantic segmentation and depth regression [48], and inverse sensor model learning [49]. However, MC dropout requires multiple forward passes during inference and is less robust to distribution shift than ensembles [50]. Moreover, MC dropout requires a dropout architecture element, which may not always be desirable. Ensembles [32] are regarded as a robust epistemic uncertainty estimation technique, but can be expensive to train and perform inference, limiting their use in robotics. One-shot evidential uncertainty estimation methods, such as deep evidential regression, have recently shown promising results in perception tasks, like depth estimation [24, 25]. In this work, we explore the use of the PostNet evidential deep learning technique [34] within a trajectory prediction task. PostNet does not require OOD data during training and only needs one forward pass during inference to obtain both the model output and the epistemic uncertainty estimate. We are interested in how this technique scales to high-dimensional data necessary for trajectory prediction.

## 3 Methods

**Problem Definition.** We consider the following trajectory prediction setting. We assume the input representation $x$ consists of a combination of the road structure (e.g., a high-definition (HD) map), the past trajectory and current state for the agent of interest, and the past trajectory information for the surrounding agents. The agent's state consists of speed $v$, acceleration $a$, and heading change rate $h$. The goal of the trajectory prediction task is to predict a 2D position vector $y \in \mathbb{R}^{2 \times T}$ for a time horizon of $T$ time steps into the future. We assume that the trajectory prediction architecture has a set of discrete anchors with ground truth labels $d \in \{1, \dots, C\}$. For example, the MultiPath [37] architecture uses k-means to cluster trajectories into a discrete latent space and the CoverNet [38] model constructs a set of possible trajectories with a specified level of coverage.

**Posterior Network (PostNet).** To estimate epistemic uncertainty in the trajectory prediction setting, we use ideas from PostNet [34]. PostNet is an evidential deep learning approach [19, 20] that uses normalizing flows [51] to learn a closed-form posterior distribution over predicted probabilities. We analyze its performance as applied to the discrete anchor space of the trajectory prediction task. PostNet's posterior distribution encompasses both aleatoric and epistemic uncertainties without requiring OOD data during training. The uncertainty is represented by a Dirichlet distribution, the conjugate prior of the categorical distribution. This approach is one-shot as it takes one network pass to compute the epistemic distribution $q^{(i)}$ and the aleatoric distribution $\bar{p}^{(i)}$ for an input $x^{(i)}$,

$$q^{(i)} = \text{Dir}(\alpha^{(i)}) \quad \text{and} \quad \bar{p}^{(i)} = \text{Cat}(\bar{\xi}^{(i)}) \quad \text{with} \quad \bar{\xi}_c^{(i)} = \mathbb{E}_{q^{(i)}}[\xi^{(i)}]_c = \frac{\alpha_c^{(i)}}{\alpha_0^{(i)}}, \qquad (1)$$

where $i$ is the dataset index, $c$ is the anchor class, $\alpha^{(i)} \in \mathbb{R}_+^C$ are the Dirichlet parameters, and $\alpha_0^{(i)} = \sum_{c=1}^{C} \alpha_c^{(i)}$ is the total amount of evidence allocated to the input (higher $\alpha_0$ indicates lower epistemic uncertainty). The parameters $\xi^{(i)} \in \left\{ [0,1]^C \mid \sum_c \xi_c^{(i)} = 1 \right\}$ of a categorical distribution

$p^{(i)} = \text{Cat}(\xi^{(i)})$ can be sampled from the epistemic distribution: $\xi^{(i)} \sim q^{(i)}$. An anchor prediction $\hat{d}^{(i)}$ is made according to: $\hat{d}^{(i)} = \arg\max_c \bar{\xi}_c^{(i)}$. The Dirichlet parameters $\alpha^{(i)}$ are constructed as $\alpha^{(i)} = \beta^{\text{prior}} + \beta^{(i)}$, where $\beta^{\text{prior}} \in \mathbb{R}_+^C$ is a fixed prior and $\beta^{(i)} \in \mathbb{R}_+^C$ represents learned pseudo-counts as evidence for an input $x^{(i)}$. Following Charpentier et al. [34], we use an uninformative prior with $\beta^{\text{prior}} = 1$. With lower confidence (smaller $\alpha$ parameters), we want the learned distribution to be close to the uninformative prior parameterized by $\beta_{\text{prior}}$. The pseudo-counts $\beta^{(i)}$ are defined as,

$$\beta_c^{(i)} = N_c \cdot r(z^{(i)} \mid c; \phi), \tag{2}$$

where $z$ is a low-dimensional continuous latent space, $\phi$ contains the network parameters, and $N_c$ reflects the ground truth count for an anchor class $c$, serving as a per class certainty budget. The probability density $r(z^{(i)} \mid c; \phi)$ is learned over the low-dimensional latent space $z$ to encourage tractability and scaling of the algorithm to high-dimensional inputs. First, a neural network encodes the input $x^{(i)}$ into the latent space, $z^{(i)} = f_\theta(x^{(i)})$. Then, due to its representational capacity, a normalizing flow is used to learn the distribution over this latent space [51, 52]. It is important that $r(z^{(i)} \mid c; \phi)$ be a normalized density to encourage the $\alpha_0^{(i)}$ parameters to be high for high-density, in-distribution (ID) regions (low epistemic uncertainty) and low for low-density, OOD regions (high epistemic uncertainty). As $r(z^{(i)} \mid c; \phi)$ goes to zero, the $\alpha^{(i)}$ parameters reduce to the uninformative prior $\beta^{\text{prior}} = 1$. To optimize PostNet, we use the evidence lower bound (ELBO) loss [53],

$$\mathcal{L}_{\text{ELBO}} = \frac{1}{N} \sum_{i=1}^{N} -\mathbb{E}_{q^{(i)}} \left[ \log p^{(i)}(d^{(i)}) \right] + \text{KL}(q^{(i)} \mid\mid \text{Dir}(1)). \tag{3}$$

**Interpretable Self-Aware Prediction (ISAP).** The PostNet approach to uncertainty quantification naturally transfers to trajectory prediction models with supervised, discrete latent anchors. However, to make the uncertainty estimates more informative in complex settings (e.g., urban scenes), we infuse interpretability into the latent space $z$, forming the proposed ISAP approach. We encode the input $x$ into three separate latent variables: $z_{\text{agent}}$, $z_{\text{map}}$, and $z_{\text{sc}}$, representing the agent's past behavior, the road structure map, and the social context surrounding the agent of interest, respectively. Such a decomposition has been shown to be effective at the input level for trajectory prediction, supporting our choice of interpretable structure [54]. These semantic concepts are encoded into the latent space using accompanying decoder components. The decoders are self-supervised with the input $x$ split into the agent's past trajectory, the road structure representation, and the past trajectories of other agents. The decoder weights are learned through associated reconstruction losses.

The ISAP network then outputs parameters for three Dirichlet distributions corresponding to the three semantic concepts. These Dirichlet distributions are combined through an equally weighted average of their parameters, signifying a uniform prior over the three categories,

$$\alpha = (\alpha_{\text{agent}} + \alpha_{\text{map}} + \alpha_{\text{sc}})/3. \tag{4}$$

These $\alpha$ parameters are used to construct the distributions in Eq. (1). The predicted trajectory is the most likely anchor trajectory according to the aleatoric categorical distribution $\bar{p}^{(i)}$. The Dirichlet distribution $q^{(i)}$ defines the epistemic uncertainty of the prediction. The full ISAP loss is then,

$$\mathcal{L} = \mathcal{L}_{\text{ELBO}} + \lambda_{\text{agent}}\mathcal{L}_{\text{rec,agent}} + \lambda_{\text{map}}\mathcal{L}_{\text{rec,map}} + \lambda_{\text{sc}}\mathcal{L}_{\text{rec,sc}}, \tag{5}$$

where $\lambda_{\text{agent}}$, $\lambda_{\text{map}}$, and $\lambda_{\text{sc}}$ are scaling coefficients for each reconstruction loss term. Additional details on the reconstruction loss terms can be found in Appendix A. Postels et al. [27, 55] demonstrate that regularization of the latent space in terms of reconstruction capability improves epistemic uncertainty estimation. These findings further support our choice of interpretable architecture for the epistemic uncertainty quantification task. The full ISAP architecture is illustrated in Fig. 1.

## 4  Experiments

We empirically validate the epistemic uncertainty estimation and OOD detection capabilities of our ISAP paradigm. All models are trained on a single NVIDIA GeForce RTX 2080 Ti GPU. Further details are provided in Appendices A and B.

**Data.** We test ISAP on the NuScenes [40] autonomous driving dataset. The predictions are made for $6\,\text{s}$ in the future based on $1\,\text{s}$ of past data collected at $2\,\text{Hz}$, following Phan-Minh et al. [38]. The

input representation $x \in [0, 1]^{500 \times 500 \times 3}$ combines the agent's past trajectory, HD map, and past trajectories of other agents into a bird's-eye view rendering of the scene (see Fig. 1). The agent's state $[v, a, h]$ serves as input to each network branch. We consider two OOD data splits. First, we split the data according to the agent's past trajectory. ID input trajectories are chosen to be slower than OOD ones. We use the $\ell_2$ distance between the oldest and most recent waypoints as a heuristic for trajectory 'speed'. We threshold the ID data to have an $\ell_2$ distance of less than $10\,\text{m}$, leaving faster trajectories for OOD data. We also consider an OOD data split according to the map structure. ID data is taken from Singapore (left-side driving) and does not contain 'roundabout' or 'big street' in the description. OOD data is from Boston (right-hand driving) with 'roundabout' in the description. Since the metadata refers to the scene and not the current local map, some straight roads exist in the OOD data as well. We verify that our chosen OOD splits are difficult to generalize to for a trajectory prediction model and, thus, important for OOD detection in Appendix C.

**Architecture Details.**   For our trajectory prediction architecture, we employ the CoverNet [38] model, which is the baseline technique for the NuScenes prediction task. This model is convenient because it frames trajectory prediction as classification over a predefined set of trajectories. Thus, we can directly integrate our ISAP approach with this architecture. For our experiments, we use a trajectory anchor set of size $64$ for classification [38]. The latent variables $z_\text{agent}$, $z_\text{map}$, and $z_\text{sc}$ are set to be four-dimensional, as this low dimensionality results in good uncertainty estimation and computational efficiency for the normalizing flows. The probability density $r(z^{(i)} \mid c; \phi)$ for each of the three latent variables is modeled with radial normalizing flows consisting of eight layers as done by Charpentier et al. [34]. The map and social context decoders take as input features one layer upstream of $z_{map}$ and $z_{sc}$ of dimension $4,096$ to enable higher representational capacity. We modify the ELBO loss to use CoverNet's constant lattice loss in the reconstruction term. The classification labels are the trajectory anchor classes with the smallest $\ell_2$ distance to the ground truth trajectories.

**Baselines.**   We consider three baselines to our approach: CoverNet [38], Post-CoverNet, and ensembles [32]. We benchmark against CoverNet for trajectory prediction and calibration performance without our modifications. Post-CoverNet is an ablation of our ISAP approach without the interpretability element, instead having a single, non-interpretable latent variable $z$. Finally, ensembles are SOTA for estimating epistemic uncertainty for neural network models. Gustafsson et al. [26] show that ensembles [32] consistently outperform MC dropout. Thus, we baseline against the more performant of the two approaches with $N = 5$ and $N = 10$ models in the ensemble.

**Metrics.**   We employ a variety of metrics to investigate how well ISAP (1) estimates epistemic uncertainty and (2) maintains trajectory prediction performance. To measure trajectory prediction performance, we use standard trajectory prediction metrics [38]: minimum average displacement error over the most likely $k$ modes ($\text{minADE}_k$) and final displacement error (FDE). Lower is better.

We then evaluate the uncertainty estimation performance. Following Charpentier et al. [34], we use the area under the receiver operating characteristic (AUROC) and average precision (APR) to compute the confidence calibration in the predictions (higher is better). We want the network to output high confidence for correct predictions (labeled 1) and low confidence for incorrect ones (labeled 0). The scores to compute the aleatoric confidence are $\max_c \bar{\xi}_c^{(i)}$. For epistemic confidence, the scores are $\max_c \alpha_c$ for Post-CoverNet and ISAP and $1/Var_c$ for ensembles where $Var_c$ is the empirical variance of the predicted class probability across the ensemble. The expected calibration error (ECE) compares the output distribution to model accuracy. The Brier score is another calibration metric: $\frac{1}{N} \sum_{i=1}^{N} \|\bar{\xi}^{(i)} - d^{(i)}\|$ where $d^{(i)}$ are one-hot labels. Lower is better for these metrics.

To evaluate OOD detection performance, we use AUROC and APR with labels 0 for OOD and 1 for ID data (higher is better). For OOD detection based on aleatoric uncertainty, the scores are $\max_c \bar{\xi}_c^{(i)}$. When based on epistemic uncertainty, the scores are $\alpha_0^{(i)}$ for Post-CoverNet and ISAP and $1/Var_c$ for ensembles. To provide further intuition for Post-CoverNet and ISAP, we also report the ratio of the average sums of the Dirichlet parameters for ID and OOD data $\bar{\alpha}_{0,OOD}/\bar{\alpha}_{0,ID}$ (lower is better). Finally, we consider entropy as an OOD detection indicator in Appendix D.

## 5   Results

**Quantitative Results.**   The trajectory prediction performance is reported in Table 1 on ID test data. As could be expected, the best performing approaches are the ensemble baselines. Ensembles tend to be more robust than their single network counterparts, in this case represented by CoverNet [38].

Table 1: Trajectory prediction metrics (lower is better) on ID test data. The best performance is highlighted in bold. Our methods perform comparably to ensembles and the original CoverNet model.

| | CoverNet [38] | Ensemble [32] (N = 5) | Ensemble [32] (N = 10) | Post-CoverNet (Ours) | ISAP (Ours) |
|---|---|---|---|---|---|
| | | **Input Past Trajectory Experiment** | | | |
| minADE$_1$ | 4.327 | **4.241** | 4.246 | 4.529 | 4.711 |
| minADE$_5$ | 1.885 | 1.867 | **1.859** | 1.951 | 2.004 |
| minADE$_{10}$ | 1.545 | **1.529** | 1.539 | 1.581 | 1.599 |
| minADE$_{15}$ | 1.413 | **1.421** | 1.423 | 1.440 | 1.474 |
| FDE | 9.474 | **9.270** | 9.293 | 10.009 | 10.177 |
| | | **Map-Based Experiment** | | | |
| minADE$_1$ | 4.732 | **4.227** | **4.227** | 4.726 | 4.822 |
| minADE$_5$ | 2.115 | 2.053 | **2.019** | 2.069 | 2.149 |
| minADE$_{10}$ | 1.731 | **1.686** | 1.689 | 1.719 | 1.737 |
| minADE$_{15}$ | 1.578 | 1.556 | **1.555** | 1.583 | 1.600 |
| FDE | 10.590 | 9.344 | **9.318** | 10.531 | 10.503 |

Interestingly, the smaller ensemble ($N = 5$) slightly outperforms the bigger ensemble ($N = 10$) for the input past trajectory experiment, indicating that the higher variability among models may cause a small drop in performance. As Post-CoverNet and ISAP add competing terms to the trajectory prediction objective, it is not surprising that the trajectory prediction performance is mildly compromised as a result. However, it is encouraging that the performance drop is only slight.

The focus of our work is on accurately estimating the epistemic uncertainty of the trajectory prediction model. The uncertainty quantification results are presented in Table 2. The ISAP model outperforms the baseline approaches across almost all metrics for the input past trajectory experiment. We make two interesting observations. The first is that ISAP outperforms Post-CoverNet in epistemic uncertainty estimation. It appears that the interpretability encoded into the latent space within ISAP helps its performance, particularly in OOD data detection. We hypothesize that distributing the uncertainty over simpler, interpretable latent variables makes the uncertainty estimation task easier. The second observation is that ISAP outperforms ensembles in OOD detection. Ensembles are often the canonical method for OOD detection; however, ISAP and Post-CoverNet outperform the smaller ensemble by a large margin. The bigger ensemble approaches Post-CoverNet performance, but ISAP still outperforms. This result supports similar findings by Charpentier et al. [34] for the PostNet architecture on smaller classification tasks.

The results for the map-based experiment largely follow the trends observed in the input past trajectory experiment. ISAP outperforms the ensembles in OOD detection. Generally, we did not find the confidence metrics or the Brier and ECE scores to be reflective of the OOD detection performance. The map-based experiment is overall more challenging than the input past trajectory experiment since the filters used to differentiate map characteristics describe scenes rather than local maps, resulting in straight roads appearing in OOD data. As such, the ID and OOD split is not as clear-cut as in the input past trajectory experiment. Nevertheless, we observe similar trends in performance across the methods for both experiments. ISAP consistently outperforms the baselines in OOD detection, while remaining performant in trajectory prediction.

We investigate how well the learned pseudo-counts $\alpha_0$ reflect the true data distribution for the input past trajectory experiment in Fig. 2. We plot the true data distribution as a histogram over the $\ell_2$ distance between the oldest and most recent waypoints in the agent's past trajectory. We distinguish between ID (green) and OOD (orange) examples. The data peaks at $0\,\mathrm{m}$ (stopped) and at around $5\,\mathrm{m}$ ($18\,\mathrm{km/h}$), and then slopes off for OOD data. The learned $\alpha_{0,\mathrm{agent}}$ parameters reflect these trends. As the ID and OOD difference is less obvious for the map and social context latent variables, the $\alpha_{0,\mathrm{map}}$ and $\alpha_{0,\mathrm{sc}}$ trends are more flat across both data types, although $\alpha_{0,\mathrm{map}}$ still shows a clear distinction. We hypothesize that the road configuration (e.g., multi-lane highway versus roundabout) is correlated with agent speed, while the social context may not always reflect the agent's speed.

**Qualitative Results.** Fig. 3 presents qualitative results for ISAP on ID and OOD examples. The figure shows the input to the network, the decoded latent variables, and their associated $\alpha_0$ values. In the input past trajectory experiment, the ID example has a slower (shorter) input trajectory than the OOD example. The ISAP network produces a clear distinction between the ID and OOD examples in the Dirichlet parameter reflecting the agent's past trajectory, as expected. The ID $\alpha_{0,\mathrm{agent}}$ value is much higher (lower uncertainty) than that for OOD. The OOD $\alpha_{0,\mathrm{agent}}$ value reaches almost total

Table 2: Uncertainty estimation metrics. If there are two numbers, they are for ID (OOD) test data. Otherwise, the data is detailed in Section 4. The best performance is in bold. Our methods outperform across most metrics.

| | CoverNet [38] | Ensemble [32] (N = 5) | Ensemble [32] (N = 10) | Post-CoverNet (Ours) | ISAP (Ours) |
|---|---|---|---|---|---|
| | | | **Input Past Trajectory Experiment** | | |
| Alea. Conf. (AUROC) ↑ | 0.638 (0.430) | 0.638 (0.399) | 0.636 (0.419) | 0.630 (0.721) | **0.652** (**0.733**) |
| Epi. Conf. (AUROC) ↑ | – | 0.455 (0.648) | 0.434 (**0.991**) | 0.573 (0.789) | **0.621** (0.745) |
| Alea. Conf. (APR) ↑ | 0.525 (0.171) | **0.551** (0.180) | 0.542 (0.179) | 0.465 (0.262) | 0.486 (**0.281**) |
| Epi. Conf. (APR) ↑ | – | 0.089 (0.001) | 0.086 (0.034) | 0.408 (0.293) | **0.451** (**0.316**) |
| ECE ↓ | 0.021 (0.339) | 0.045 (0.317) | 0.056 (0.280) | **0.017** (0.198) | 0.048 (**0.053**) |
| Brier Score ↓ | 0.837 (1.011) | **0.835** (1.007) | 0.840 (1.000) | 0.857 (0.963) | 0.850 (**0.960**) |
| Alea. OOD (APR) ↑ | 0.542 | 0.530 | 0.538 | 0.833 | **0.930** |
| Epi. OOD (APR) ↑ | – | 0.810 | 0.961 | 0.960 | **0.976** |
| Alea. OOD (AUROC) ↑ | 0.241 | 0.218 | 0.240 | 0.652 | **0.871** |
| Epi. OOD (AUROC) ↑ | – | 0.693 | 0.913 | 0.919 | **0.955** |
| $\bar{\alpha}_{0,OOD}/\bar{\alpha}_{0,ID}$ ↓ | – | – | – | 0.171 | **0.145** |
| | | | **Map-Based Experiment** | | |
| Alea. Conf. (AUROC) ↑ | 0.594 (0.415) | 0.629 (**0.636**) | **0.631** (0.616) | 0.610 (0.593) | 0.582 (0.630) |
| Epi. Conf. (AUROC) ↑ | – | 0.582 (0.618) | **0.635** (0.681) | 0.610 (0.647) | 0.575 (**0.707**) |
| Alea. Conf. (APR) ↑ | 0.399 (0.126) | **0.531** (0.225) | 0.525 (**0.292**) | 0.452 (0.222) | 0.428 (0.187) |
| Epi. Conf. (APR) ↑ | – | 0.103 (0.043) | 0.129 (0.164) | **0.463** (**0.284**) | 0.425 (0.281) |
| ECE ↓ | 0.056 (0.200) | 0.080 (0.118) | 0.092 (**0.087**) | **0.046** (0.132) | 0.113 (0.102) |
| Brier Score ↓ | 0.873 (0.985) | **0.845** (0.964) | 0.852 (**0.949**) | 0.871 (0.973) | 0.868 (0.968) |
| Alea. OOD (APR) ↑ | 0.906 | 0.946 | 0.941 | 0.913 | **0.956** |
| Epi. OOD (APR) ↑ | – | 0.876 | 0.875 | 0.941 | **0.968** |
| Alea. OOD (AUROC) ↑ | 0.690 | 0.786 | 0.777 | 0.724 | **0.806** |
| Epi. OOD (AUROC) ↑ | – | 0.552 | 0.553 | 0.756 | **0.838** |
| $\bar{\alpha}_{0,OOD}/\bar{\alpha}_{0,ID}$ ↓ | – | – | – | 0.502 | **0.245** |

uncertainty as the uniform prior over 64 possible anchors would produce $\alpha_0 = 64$. The $\alpha_{0,\mathrm{map}}$ and $\alpha_{0,\mathrm{sc}}$ values are also lower for the OOD input than the ID one. We hypothesize that intersections are more likely in the ID data, where the agent of interest is traveling at a slow speed, than large four-lane roads that resemble highways as in the OOD example. A surprising observation in Fig. 3 is that the OOD $\alpha_{0,\mathrm{sc}}$ value is quite high, suggesting low epistemic uncertainty. In both the ID and OOD examples, the agent of interest is traveling in traffic with many cars on the road. Such scenarios are likely common for the slow-speed training data, indicating ID characteristics. Thus, our ISAP paradigm provides insight into the interpretable sources of epistemic uncertainty.

In the map-based experiment, the ID agent of interest is traveling at a moderate speed along a straight, two-lane road with a car ahead. This scenario is pretty likely according to our expectations and the learned $\alpha_0$ pseudocounts. In the OOD data input, the map shows a roundabout, which should not be present in the ID data. As expected, the OOD $\alpha_{0,\mathrm{map}}$ value is significantly lower than that of the ID example as the network has not seen this intersection type during training.

## 6 Conclusions

In this paper, we propose a new trajectory prediction paradigm called Interpretable Self-Aware Prediction (ISAP). ISAP learns the aleatoric and epistemic uncertainty over a discrete set of supervised anchors in a trajectory prediction architecture. These uncertainties are estimated in one-shot by using ideas from evidential deep learning. We introduce interpretability into the epistemic uncertainty

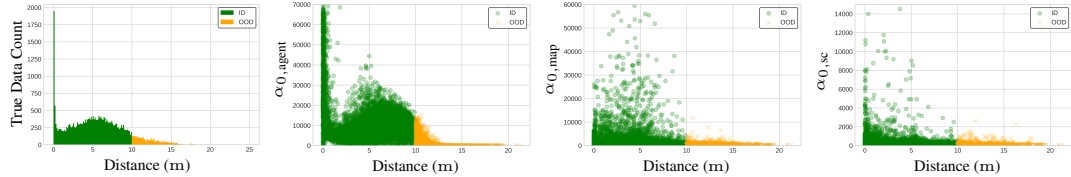

Figure 2: Learned $\alpha_{0,\mathrm{agent}}$, $\alpha_{0,\mathrm{map}}$, and $\alpha_{0,\mathrm{sc}}$ pseudocounts (3 right plots) in comparison to a histogram of the true data counts (left plot). All data is from the ID and OOD test sets, aside from the histogram ID data, which reflects the training distribution. Outliers with large $\alpha_0$ values are omitted for visualization. The $\ell_2$ distance (in meters) between the oldest and most recent 2D waypoints is a proxy for input trajectory speed. The $\alpha_{0,\mathrm{agent}}$ parameters accurately reflect the trends in the true data distribution with increasing input trajectory speed.

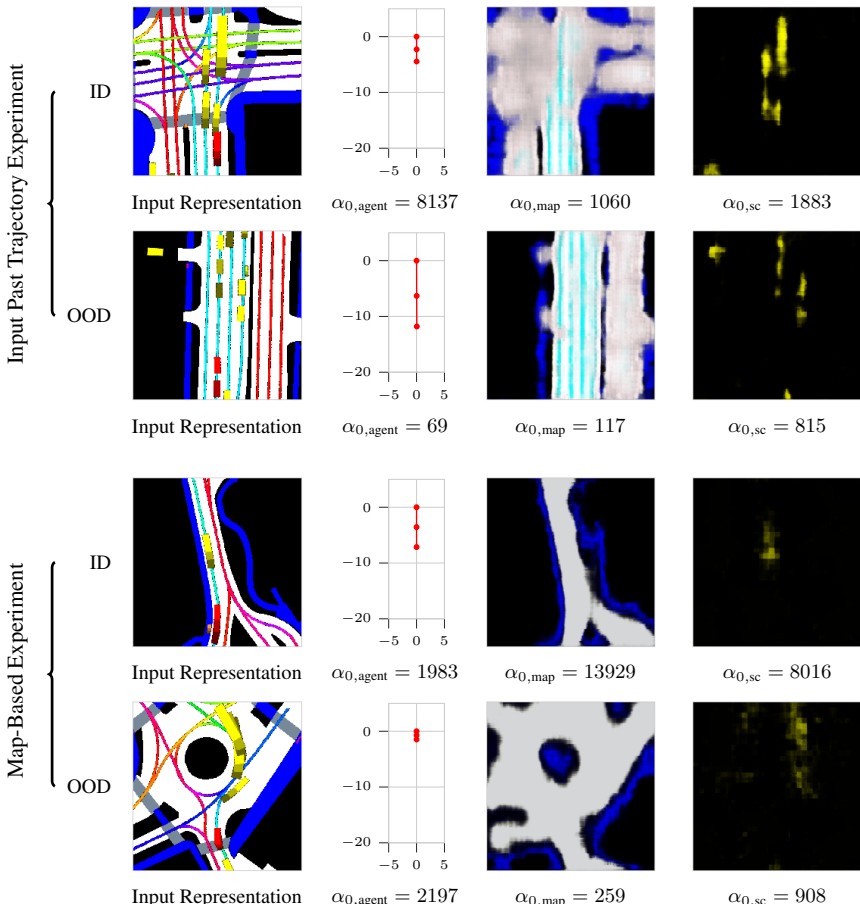

Figure 3: Qualitative results for our ISAP framework on ID and OOD examples. Plot axes for the decoded input trajectories are in meters. In the input past trajectory experiment, the OOD example has a faster input trajectory than the ID one. Thus, the learned epistemic uncertainty in the agent behavior latent variable is higher ($\alpha_{0,\text{agent}}$ is lower) for the OOD case than the ID case. In the map-based experiment, the OOD example portrays a roundabout, while the ID one shows a straight, two-lane road. Roundabouts should not be present in the ID data. Thus, the learned epistemic uncertainty in the map latent variable is higher ($\alpha_{0,\text{map}}$ is lower) for the OOD case than the ID case.

estimate by subdividing the uncertainty into the semantic concepts: past agent behavior, road structure map, and social context. Our approach maintains comparable trajectory prediction performance to an unmodified trajectory prediction architecture and outperforms established techniques, like ensembles, in uncertainty estimation while requiring only a single network pass during inference.

**Limitations.** Although we show that normalizing flows in the ISAP framework are performant for uncertainty estimation, they can be brittle and slow to train. They also struggle to scale to large latent spaces, restricting the representational capacity of the latent space. Moreover, by design, we impose an inductive bias on our ISAP framework for trajectory prediction in the context of AVs. For other applications, like assistive robotics, the interpretable structure may need to be adapted (e.g., to include task-level concepts, such as cooking or cleaning). Lastly, although the drop in trajectory prediction metrics is small with the addition of epistemic uncertainty estimation, it is prudent to consider how this gap could be closed in future work.

**Future Work.** Another promising avenue for future work is extending our ISAP paradigm to map-centric environment prediction [56–58]. In map-centric prediction, the inputs and outputs are sequences of occupancy grids, which are higher dimensional than inputs and outputs in traditional, object-centric trajectory prediction. Map-centric representations are robust to partial occlusions, can handle an arbitrary number of agents in the scene, and do not require significant preprocessing. Thus, scaling epistemic uncertainty estimation to these settings is an exciting open research problem.

**Acknowledgments**

This work was supported by funding from Waymo. We thank Ben Sapp and Dragomir Anguelov for insightful discussions throughout the project. We thank Spencer M. Richards and Ransalu Senanayake for their invaluable feedback.

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
