# OpenReview forum: "Interpretable Self-Aware Neural Networks for Robust Trajectory Prediction"
_robot-learning.org/CoRL/2022/Conference — CoRL 2022 Poster_

### Official Review · Reviewer_UZiy · 2022-07-08

**Originality:** Good
**Technical Quality:** Good
**Clarity Of Presentation:** Very Good
**Impact:** 3

**Recommendation:**

Weak Accept: I recommend accepting the paper, but will not argue for my recommendation if the majority of other reviewers have a different opinion.

**Summary:**

An uncertainty aware trajectory prediction framework of other road participants is presented, showing similar accuracy to deterministic baselines but improved calibration and out-of-distribution detection capabilities.

Both model (epistemic) and data (aleatoric) uncertainty are estimated and the former is further separated into coming from three sources, namely the agent behavior, the social context and the road configuration.

As fast inference is important in the autonomous driving setting, the uncertainty is obtained through a single forward pass using ideas from the evidential deep learning literature, i.e. it is modeled using a learned Dirichlet distribution.

**Issues:**

1. Report common calibration metrics like expected calibration error.
2. Report confidence in wrong predictions.
3. Improve exposition of in- and out-of-distribution results.
4. Properly justify the choice of uncertainty decomposing factors.
5. Make use of stronger and more realistic out-of-distribution data.
6. Rethink the use of "self-awareness".
7. Fix typos.

**Quality Of The Limitations Section:**

Additional details required

**Reviewer Expertise:**

3: The reviewer is fairly confident that the evaluation is correct

**Robotics Focus:**

Highly relevant to robotics but no hardware experiments

**Strengths And Weaknesses:**

## Strength
* Uncertainty estimation is important for safety critical applications like autonomous driving
* Fast, i.e. suitable to the autonomous driving setting
* Only slight drop in accuracy compared to deterministic and ensemble baselines
* Evaluated based on calibration and in- and out-of-distribution separation
* Increased interpretability through decomposition into three sources of uncertainty
* Clean exposition

## Weaknesses
* Calibration is measured using the Brier score instead of the expected calibration error which is commonly used.
* Confidence in wrong predictions is not included which might arguably be of even greater importance than confidence in correct ones in safety critical contexts.
* In- and out-of-distribution separation could be visualized and exploited using predictive entropy.
* Choice of uncertainty decomposing variables is not well justified.
* Out-of-distribution is defined through (arbitrary) speed threshold. Corruption, adversarial attacks or other datasets might yield stronger results.
* Describing the system as "self-aware" seems a bit exaggerated and out of place.
* Typos:
  * "may be not be" (line 4)
  * $\lambda_{agent}$, $\lambda_{agent}$ and $\lambda_{agent}$ (line 163)

**Summary Of Recommendation:**

The paper shows promising advances for much needed uncertainty estimation in safety-critical, real-time systems. The decomposition of the epistemic uncertainty into different sources and the adaption of evidential deep learning approaches to trajectory prediction are interesting additions to the existing literature. The results are promising but not entirely convincing.

---

> ### Author Response · Authors · 2022-08-19
> **Response to Reviewer UZiy**
>
> **Comment:**
>
> We thank the reviewer for their thoughtful comments and suggestions. We were happy to see that the reviewer saw a number of strengths in our work, including the importance of the considered problem, evaluation thoroughness, interpretability, and clean exposition. We address the reviewer comments below.
>
> ### Weaknesses:
>
> > Calibration is measured using the Brier score instead of the expected calibration error which is commonly used.
>
> Thank you for the suggestion! We were following the uncertainty evaluation scheme in PostNet [34]. We have now also added the expected calibration error (ECE) metric. We include **in attachment** the updated Table 2. For the input past trajectory experiment and OOD data (denoted in parentheses), we found the expected calibration error to be lowest for ISAP. However, we would like to highlight that the main goal of our approach is to detect OOD examples based on model outputs. Thus, the most critical metrics are the aleatoric and epistemic OOD detection using AUROC and APR (lines 7-10 in the new table).
>
> > Confidence in wrong predictions is not included which might arguably be of even greater importance than confidence in correct ones in safety critical contexts.
>
> We believe there may be some confusion here. The confidence metric measures how well the confidence is calibrated both for correct and incorrect predictions. Specifically, we want the network to output high confidence for correctly predicted examples and low confidence for incorrect predictions. We achieve a measure for this calibration AUROC and APR with aleatoric and epistemic confidence scores. We have updated the wording in lines 217-222 accordingly to make this clear.
>
> > In- and out-of-distribution separation could be visualized and exploited using predictive entropy.
>
> Thank you for the suggestion! We generally follow the OOD detection metrics used in PostNet [34]. We have included entropy histograms for ID and OOD data on our ISAP approach and ensembles (N = 10) for the input past trajectory and map-based experiments **in attachment**. In both experiments, ISAP provided a more clear distinction between ID and OOD data in terms of entropy than the ensemble. We will include these plots and corresponding discussion in the appendix.
>
> > Choice of uncertainty decomposing variables is not well justified.
>
> Our choice of agent past behavior, map, and social context decomposition is supported by successful use of a similar decomposition as input to a trajectory prediction model [54] (lines 151-155). Prior work [55-27] has also shown that adding reconstruction terms to regularize the latent space can be helpful for epistemic uncertainty estimation (lines 164-166). We are not sure if the reviewer is looking for a different type of justification here?
>
> > Out-of-distribution is defined through (arbitrary) speed threshold. Corruption, adversarial attacks or other datasets might yield stronger results.
>
> We gave significant thought to the choice of OOD split in the data. Since the uncertainty estimation we conduct in ISAP is task-aware, it was important that the OOD split affected trajectory prediction performance. The relative speed of the trajectory provided a significant difference in trajectory prediction for the CoverNet baseline between the two data categories as shown in the table included below on ID (OOD) data.
>
> Furthermore, the $\ell_2$ past trajectory distance is a continuous measure which allows us to see how the \alpha values evolve with the increase in $\ell_2$ in Fig. 2. ISAP’s ability to clearly distinguish between the two data categories (see quantitative results in Table 2) further supports this choice of data split.
>
> In addition, we consider a map-based split in Appendix B, where we keep left-side driving in Singapore and avoid big streets and roundabouts for in distribution data. For OOD, we use data from Boston (right-side driving) and ensure ‘roundabout’ is in the scene description (lines 508-511). We are currently incorporating this experiment into the main paper.
>
> With these two experiments, our goal was to demonstrate the usefulness of the interpretability element to our proposed ISAP approach.
> While we could use corruption or adversarial attacks to achieve OOD examples, we were looking to validate our OOD approach on naturally occurring and more plausible OOD data.
>
> |Experiment | minADE$_{1}$      |  FDE
> |------------ | ----------- | ----------- |
> Input past trajectory    |  4.327 (7.130)    |  9.474 (13.632)
> Map | 4.732 (6.111) | 10.590 (13.464)
>
> > Describing the system as "self-aware" seems a bit exaggerated and out of place.
>
> By the term 'self-aware', we mean that the ISAP model tracks its epistemic uncertainty. We define this term in the paper in lines 3-5 and again in lines 21-27 to avoid ambiguity.
>
> ### Issues
>
> We address the issues listed by the reviewer above. Thank you to the reviewer for catching the two typos! We have fixed them in the paper.
>
> **Zip File:**
>
> /attachment/57ef021c644660eab0c986f8632ed404b7321c84.zip

---

### Official Review · Reviewer_we25 · 2022-07-27

**Originality:** Good
**Technical Quality:** Very Good
**Clarity Of Presentation:** Very Good
**Impact:** 3

**Recommendation:**

Weak Accept: I recommend accepting the paper, but will not argue for my recommendation if the majority of other reviewers have a different opinion.

**Summary:**

This paper proposes a framework, named ISAP, for trajectory prediction with interpretable uncertainty estimates. The key idea enabling the interpretability is to split the network architecture into three low-dimensional but general components in the latent space: agents past behaviour, map, and social context. By using normalizing flows, the technique the paper proposes does not require OOD data for training. ISAP is benchmarked against nuScenes where in-distribution data is defined by a measure of the speed of the trajectory. The results show comparable trajectory prediction with baselines with improved uncertainty estimation.

**Issues:**

1. Scores for various metrics, even the more important ones look very similar across the baselines.
1. The limitations mentioned on network capacity and difficulty of training normalizing flows seem like relatively small and engineering-focused limitations, especially since most techniques for these high-dimensional datasets that include an uncertainty estimate are already difficult to train. To what extent are the discrete anchors generalizable/scalable? Another significant limitation is that ISAP is a pure prediction method.
1. Helpful additional information in the appendix. Some language can be made more specific/technical.


**Quality Of The Limitations Section:**

Limitations are addressed clearly

**Reviewer Expertise:**

3: The reviewer is fairly confident that the evaluation is correct

**Robotics Focus:**

Relevant but unlikely to deploy to hardware in near future

**Strengths And Weaknesses:**

Strengths

1. Algorithm (ISAP) is evaluated on a subset of the nuScenes dataset, which is relatively standard and contains a wide variety of conditions to test uncertainty over the latent space parameters in the method.
2. Approach is described precisely in a way that seems possible to reproduce. Although I did not run the code, it looks well documented and provides clear instructions for running the experiments in the paper. The description of PostNet in particular was succinct but informative.
3. Good selection of competitive baselines that the algorithm is compared against across a wide variety of metrics for uncertainty estimation.
3. Experiments show competitive trajectory prediction and superior uncertainty estimation compared to baselines.
4. Qualitative experiments that show relative magnitudes of the learned uncertainty parameters along with thoughtful discussions interpreting those qualitative results in the context of the data.
5. The paper is well-written and presented. The figures use space well to illustrate concepts and ground them in the data used.
6. Very thorough discussion of related work covering a wide range of papers
7. Uses road structure as another way to detect OOD data, which is much harder to capture in a straightforward way from the state space.

Weaknesses

1. The biggest weakness in my opinion is the definitions used of OOD and ID data for evaluation. The main difference that is evaluated on is trajectory speed, which is highly relevant for trajectory prediction, but seems relatively easily to detect when velocity is part of the state space. Detecting road structure (roundabout/side of road) is a large improvement. One more would be nice, especially since the baselines seem to perform quite well.
2. The method proposed seems to be very similar to PostNet with the only difference being additional losses to form ISAP (the name for the proposed method) from PostNet. However, the majority of the paper is focused on how to split uncertainty into these semantically meaningful categories and ensuring that the benefits of that split are reflected in the results. In the rebuttal, the reviewers justified why this paper is a significant improvement on the PostNet backbone,
4. Discrete anchors with a predefined number of labels, which I don’t think is a major weakness because discretization often leads to nice properties for learned distributions, and there are many existing methods to do this effectively in a self-supervised manner.


**Summary Of Recommendation:**

The paper presents a method that is generally useful in driving domains, where being able to detect epistemic uncertainty is useful for planning.  Having 3 relatively general-purpose interpretable measures of uncertainty that have good quantitative results and sensible qualitative results is useful. The revision clarified some of the technical details and presented some results on a more meaningful OOD/ID axis on road structure. A reviewer who is more familiar with typical AUROC scores in this domain would be better suited to comment, but the differences in scores in the map-based experiment look very similar.

---

> ### Author Response · Authors · 2022-08-19
> **Response to Reviewer we25 (1/N)**
>
> We thank the reviewer for their thoughtful comments and suggestions. We were happy to see that the reviewer saw a number of strengths in our work, including the precision of our approach description, good baseline selection, superior uncertainty estimation performance, and thorough related works discussion. We were also pleased that the reviewer found our paper to be well-written and presented, particularly in our use of figures! We address the reviewer comments below.
>
> ### Weaknesses
>
> 1. > The biggest weakness in my opinion is the way the paper defined OOD and ID data in the main body of the paper, which is according to a trajectory “speed” metric which seems easy to learn because the velocity is part of the agent space. A more interesting and meaningful way to segment the data would be something such as weather conditions, city, prevalent traffic patterns, and pedestrians. The Appendix Section B shows results when the maps were from different cities, which are not as impressive as for speed, but in my opinion, warrant being in the main body of the paper since the experiment is more informative about the ability of the method to distinguish OOD/ID data on more informative (metaphorical) axes.
>
> We gave significant thought to the choice of OOD split in the data. Since the uncertainty estimation we conduct in ISAP is task-aware, it was important that the OOD split affected trajectory prediction performance. The relative speed of the trajectory provided a significant difference in trajectory prediction for the CoverNet baseline between the two data categories as shown in the table included below on ID (OOD) data. Similarly, the map experiment provided a smaller, yet still significant performance difference as well.
>
> Furthermore, the $\ell_2$ past trajectory distance is a continuous measure which allows us to see how the \alpha values evolve with the increase in $\ell_2$ in Fig. 2. ISAP’s ability to clearly distinguish between the two data categories (see quantitative results in Table 2) further supports this choice of data split.
>
> As suggested by the reviewer, we will also move our map-based experiment into the main paper. With these two experiments, our goal was to also demonstrate the usefulness of the interpretability element within our proposed ISAP framework.
>
> |Experiment | minADE$_{1}$      |  FDE
> |------------ | ----------- | ----------- |
> Input past trajectory    |  4.327 (7.130)    |  9.474 (13.632)
> Map | 4.732 (6.111) | 10.590 (13.464)
>
> 2. > The method proposed seems to be very similar to PostNet with the only difference being additional losses to form ISAP (the name for the proposed method) from PostNet. However, the majority of the paper is focused on how to split uncertainty into these semantically meaningful categories and ensuring that the benefits of that split are reflected in the results.
>
> The goals for ISAP were to achieve one-shot epistemic uncertainty estimation working towards real-time capabilities and ensure that the resulting epistemic uncertainty is usable and interpretable by other autonomy stack components and engineers for further model development. These characteristics, to our knowledge, have not yet been achieved in epistemic uncertainty estimation for trajectory prediction, particularly as this is still an early research direction.
>
> Evidential deep learning methods, the class of approaches that PostNet belongs to, are able to estimate epistemic uncertainty in one-shot, and PostNet furthermore does not require explicit OOD data during training. These attributes make PostNet a suitable backbone to build upon for our ISAP approach. PostNet [34] was originally only validated on small classification tasks (e.g., tabular data, MNIST, CIFAR10), while trajectory prediction is a much more challenging, large-scale problem for which PostNet has not previously been explored. Our novelty lies in recognizing that to make the approach applicable and useful to the trajectory prediction task, we need to infuse interpretability into the estimated uncertainty, forming the ISAP framework. As the reviewer indicates, we empirically demonstrate that our proposed semantic breakdown of the uncertainty in the trajectory prediction task improves ISAP’s ability to identify OOD examples.

---

> > ### Author Response · Authors · 2022-08-19
> > **Response to Reviewer we25 (2/N)**
> >
> > 3. > What exactly does “our approach without the interpretability element.” on Line 207 mean? An ablation test that would help support the hypothesis the authors state that “distributing the uncertainty over simpler, interpretable latent variables makes the uncertainty estimation task easier” would be useful if it distributed the uncertainty over some arbitrary low-dimensional latent variables. Is Post-CoverNet doing this?
> >
> > Yes! Post-CoverNet does not break down the latent space into the three interpretable categories, but rather just uses an arbitrary latent $z$ encoding. There are no decoders present in Post-CoverNet. We have added a clarifying sentence emphasizing the differences between ISAP and Post-CoverNet in line 207: "Specifically, in Post-CoverNet, there is no separation of the latent state into three components, and, instead, Post-CoverNet has a single, non-interpretable latent state $z$."
> >
> > 5. > I found the sentence describing the distance in Figure 2 hard to understand and vague. The distance between the earliest and most recent waypoints in the agent’s past trajectory: how many of each of these points? And which distance? Distances between 2 sets of points can be defined in many ways. What is the unit? “m” suggests meters but then there seems to be a conversion between m to km/h which is confusing because those are completely different units, and ambiguous because m is used as the distance unit, and h as the heading change rate unit.
> >
> > As part of the input to the network, we consider the past trajectory of the agent of interest. We take the first (earliest) $x$-$y$ point in this trajectory and the last (latest) $x$-$y$ point and measure the $\ell_2$ distance between them. This spatial distance is measured in meters (m). This is the same distance we use to split the OOD/ID data as described in lines 179-182.
> >
> > We have rephrased the sentence in the Fig. 2 caption as follows: “The speed is represented as the $\ell_2$ distance (in meters) between the earliest and most recent $x$-$y$ waypoints in the input past trajectory.”.
> >
> > We are able to convert between this distance (in meters) and an estimated speed in km/h as the past trajectories are over 1 second of time (lines 173-174). We added in the estimated speed in km/h for some intuition, but we are happy to add a clarifying sentence there as well.
> >
> > ### Issues
> >
> > 1. > Paper warrants a more thorough discussion of the implications of the OOD/ID separation based on “trajectory speed” rather than other factors or a mix of factors that would make data out of distribution.
> >
> > Please see our response above.
> >
> > 2. > The less trivial distinctions between ID and OOD data are in the appendix rather than in the main paper. I found the experiments in the appendix much more compelling, so consider moving those into the main body of the paper.
> >
> > We are currently working on incorporating the map-based experiment into the main paper. Thank you for the suggestion!
> >
> > 3. > The limitations mentioned on network capacity and difficulty of training normalizing flows seem like relatively small and engineering-focused limitations, especially since most techniques for these high-dimensional datasets that include an uncertainty estimate are already difficult to train. For instance, what assumptions are made about the input data? ISAP seems quite specific to driving for example. To what extent are the discrete anchors generalizable/scalable? Another significant limitation is that ISAP is a pure prediction method.
> >
> > Yes, you are absolutely correct. We design ISAP with trajectory prediction in the context of AVs in mind. If we were to consider trajectory prediction in an indoor environment with a mobile robot, a lot of the same semantic splits could be considered. However, for an application like assistive robotics, a different semantic breakdown might make more sense that would focus on task-level concepts (e.g., cooking, cleaning) and types of rooms (e.g., kitchen, living room, bathroom). We have added this intentional bias to our limitations.
> >
> > The discrete trajectory anchors that we use were inherited from CoverNet [38]. Using discrete anchors in trajectory prediction seems to be a popular and successful approach [37-39].
> >
> > 4. > Define $\mathcal{L}_{\text{rec,agent}}$ and the other categories explicitly. If you need extra space, some of the implementation details can be moved to the appendix.
> >
> > We describe in detail the reconstruction losses in the appendix in lines 486-496. Please let us know if you are looking for something beyond this description.

---

> > > ### Author Response · Authors · 2022-08-20
> > > **Response to Reviewer we25 (3/N)**
> > >
> > > 5. > Figure 2 can be more specifically described in the first sentence of the caption to make it clear that the left shows the true data count and the right are the pseudocounts. It took a while to recognize, especially since the text is so small.
> > >
> > > We have updated the first sentence in the caption of Fig. 2 to:  "Learned $\alpha_{0, \text{agent}}$, $\alpha_{0, \text{map}}$, and $\alpha_{0, \text{sc}}$ pseudo-counts (3 right plots) in comparison to a histogram of the true data counts (left plot).". Please let us know if this helps!
> > >
> > > 5. > Furthermore, please describe the distance on the x-axis more precisely in the text and address ambiguities pointed out in the corresponding bullet point in “Strengths and Weaknesses”.
> > >
> > > Please see our response above.
> > >
> > > 6. > Minor typos: Input Represenation in Figure 3
> > >
> > > Thank you for the catch, we have fixed this typo!

---

### Official Review · Reviewer_WAfv · 2022-07-28

**Originality:** Good
**Technical Quality:** Good
**Clarity Of Presentation:** Good
**Impact:** 3

**Recommendation:**

Weak Accept: I recommend accepting the paper, but will not argue for my recommendation if the majority of other reviewers have a different opinion.

**Summary:**

This paper proposes a deep-learning based method to estimate epistemic uncertainty over a low-dimensional, interpretable latent space used for trajectory prediction. The key idea of the approach is to map the trajectory prediction data into three interpretable components: the agent, the map, and the social context and obtain a latent state encoding each of these aspects individually. To capture the certainty of each latent state, the proposed neural network outputs parameters of three Dirichlet distributions (one for the agent, the map, and the social context) that are averaged together for a total uncertainty score. Empirical results with the nuScenes dataset (where in-distribution data is slow driving vs. out-of-distribution data is fast driving) show that the proposed approach performs slightly worse than ensemble or alternative baselines on prediction metrics, but outperforms the baselines on uncertainty estimation metrics.


**Issues:**

Section 2:

–Figure 1: There are several aspects of this figure that could be improved to help the reader understand the method. First, the inputs $[v,a,h]$ are never described in the figure nor in the text. It could also help to label the input representation with x, as is referred to in the text. Second, it would be helpful to explicitly highlight where the latent states $z_{agent}$, $z_{map}$, and $z_{sc}$ are obtained in the pipeline, since these are key to the contribution. Additionally, the block called “Backbone” could be more descriptive. Finally, it could be clarified where the $\alpha$ Dirichlet parameters come from, since this is another aspect of the contribution. It is also unclear why there are only three predictions as output of the final Dir block? Are these just the three most likely trajectories, or does the method only allow for three predictions? Overall, making the text notation consistent and explicit in Figure 1 would help this figure clarify + enhance the technical content.

Section 3:

–In the summary of PostNet, it was also difficult to understand what the index (i) stands for. Please clarify this.

–In equation (4) why is it desirable to always equally weigh the three alpha parameters? Providing some intuition here would be helpful.

Section 4:

–Please clarify what each aspect of agent state means: $[v, a, h]$. Also, is this just an instantaneous state or do you obtain the states over time as well?

–Are these “types” of OOD data something that is typically a problem for predictors? It would be more powerful to first present the key “types” of OOD data that often mess up ensembles or other baselines. This would make the types of data used for testing much more convincing. Otherwise, right now it seems like somewhat of an ad-hoc choice to say that in-distribution data is “slow driving” while OOD data is “fast driving” when perhaps these types of variations in driving behavior are seen so frequently and in such large quantities in typical datasets, that SOTA predictors don’t struggle with them.

–Are the latent states $z$ assumed to be discrete or continuous? It would be helpful to clarify this.

–”Post-CoverNet is an ablation….without the interpretability element” does this just mean that there is no separation of the latent state into three components, and instead Post-CoverNet just produces a single latent state z? It would be helpful to clarify this.

–”The ground truth labels are 0 for incorrect predictions and 1 for correct predictions.” It seems like the labels shouldn’t correspond to state prediction accuracy but whether or not a data point is ID or OOD. It would be helpful to clarify this in the text.

–What does subscript $c$ refer to? What are the possible anchor classes?

–In the Brier score definition, $\bar{p}$ and $d$ seem to be incompatible: $\bar{p}$ is defined as a distribution while $d$ is a class label. Also, what kind of classes can $d$ represent? Are classes the type of agents (e.g., car, bike, truck etc.) or something else?

Section 5:

–Figure 2: It was difficult to compare the leftmost plot (whose y-axis is total data count) vs. the right three plots (whose y-axis is the $\alpha_{0, agent}$) since they have different y-axis, but look as though all four subplots should be compared equally. It would be helpful to explain how to compare / interpret these plots, or make the axes all have the same units.

–Figure 3: It’s difficult to interpret the second plots from the left (with $\alpha_{0, agent}$). Is this a spatial visualization (so the x and y axis represent the x and y of the world) where the red line is some sort of prediction? Or do the axes represent something different?

–Since the trajectory prediction for the proposed method seems to be on-par with the SOTA methods, it seems to imply that all the predictors can do a decent job of predicting the OOD examples too. So, even though the proposed method can detect OOD examples, the ultimate prediction performance (which would then affect the autonomous vehicle decision-making) seems to not need any OOD detection to achieve good ADE/FDE scores. I think these results warrant (1) investigation into “harder” OOD datasets that clearly showcase what SOTA predictors would suffer in prediction performance and (2) how being able to do OOD detection can be integrated with robot decision-making to achieve safer behaviors than what is achieved *without* OOD detection.


**Quality Of The Limitations Section:**

Additional details required

**Reviewer Expertise:**

4: The reviewer is confident but not absolutely certain that the evaluation is correct

**Robotics Focus:**

Highly relevant to robotics but no hardware experiments

**Strengths And Weaknesses:**

Strengths:
----------------
–Important and timely problem choice

–Interesting approach to balancing interpretability of the latent parameters, prediction performance, and epistemic uncertainty estimation

–Evaluation with nuScenes dataset of real highway driving on two examples: datasets where (1) ID data = slow driving vs. OOD data = fast driving and (2) ID data = left-side driving + no roundabouts + small streets vs. OOD data = right-side driving + roundabouts.

Weaknesses:
------------------
–While at a high level the motivation of this work is good, the specifics of why this method enables better prediction and ultimately, better autonomous vehicle behavior, could be significantly strengthened. From the results, it seems like predictive performance is good for the baseline predictors (even marginally better than the proposed method) which would indicate that the baselines may be successful at prediction even on the presented OOD data (and therefore enabling downstream robot behavior to be safe  /effective). This implies that there may be marginal benefits to doing OOD detection with the proposed method for the proposed scenarios. Intuitively, I’d imagine that if the predictor is presented with truly OOD data that is really difficult for it to generalize from the training data, then the prediction performance would significantly suffer. This suggests that perhaps this ID/OOD data split isn’t the best to showcase why measuring epistemic uncertainty is so necessary. For example, I’d posit that slow vs. fast variations in driving behavior are easy to generalize between and seen so frequently / in such large quantities in typical datasets, that SOTA predictors don’t struggle with speed of driving. However, OOD data like drunk drivers, U-Turns, construction zones, etc. intuitively seem like they would be very hard for SOTA predictors to deal with, thereby resulting in poor predictive performance and motivating the need for OOD detection. This paper would be strengthened by highlighting (1) ID/OOD data that SOTA predictors truly struggle with and (2) demonstrating how the proposed OOD method can enable, for example, more “safe” predictions by incorporating robot decision-making that takes safe recourse when presented with an OOD scenario.

–In general, it is challenging to understand the connection between the input $x$, the latent states $z$, and the generation of the dirichlet parameters $\alpha$ (which are the crux of the uncertainty quantification method). This makes it difficult to deeply understand the key idea of this work and its strengths/weaknesses.

–Overall, the summary of PostNet was quite challenging to follow for a reader who is not deeply familiar with this method. This makes it difficult to appreciate the contribution that hinges on this method. I would suggest changing the flow and adding some more exposition to clarify the key connections between all aspects of this predictor. For example, it would be helpful to start with how PostNet takes as input $x$ (and provide intuition for what this input is – e.g., this is a top-down RGB image of the scene) and this input is converted into a single latent state $z$ via a neural network: $z = f_\theta(x)$. Then, how does this latent state then get converted into trajectory predictions? The text never clearly states this. Then, you can mention how to obtain uncertainty estimates: PostNet starts by learning a distribution over the latent state, conditioned on a discrete anchor class, c. Here it would be very helpful to provide intuition for why this anchor class is needed – for example, what does it represent in trajectory prediction? Then, you can mention that the shape of this distribution is key to obtaining an uncertainty estimate. The epistemic uncertainty is modeled as a Dirichelt distribution with parameters alpha where $\alpha = \beta^{prior} + \beta_c$. Here is where you leverage the distribution over the latent state: $\beta_c = N_c \cdot r(z | c, \phi)$. As the probability of a latent state given the anchor class goes to zero, then $\alpha$ approaches $\beta^{prior}$. Here it would be helpful to provide intuition for why this is a desirable property. Finally, you can describe how you also model the aleatoric uncertainty via the categorical distribution and why this works / is desirable. Then, the ELBO loss can be clearly introduced as the method for how this network is trained. It would also be helpful to provide some intuition for this loss as well.


**Summary Of Recommendation:**

Although the topic of this work is timely and relevant, the proposed approach warrants deeper investigation to present a strong result. Since the trajectory prediction for the proposed method seems to be on-par with the SOTA methods, it seems to imply that all the predictors can do a decent job of predicting the OOD examples too. So, even though the proposed method can detect OOD examples, the ultimate prediction performance (which would then affect the autonomous vehicle decision-making) seems to not need any OOD detection to achieve good ADE/FDE scores. I think these results warrant (1) investigation into “harder” OOD datasets that clearly showcase what SOTA predictors would suffer in prediction performance and (2) how being able to do OOD detection can be integrated with robot decision-making to achieve safer behaviors than what is achieved without OOD detection. Due to these reasons and the current state of the manuscript, I recommend weak reject.

=====================

After discussion, I change to "weak accept".

---

> ### Author Response · Authors · 2022-08-19
> **Response to Reviewer WAfv (1/N)**
>
> We thank the reviewer for their thoughtful comments and suggestions. We were happy to see that the reviewer found our problem choice to be ‘important and timely’ and our ISAP approach to be interesting in balancing interpretability, prediction performance, and uncertainty estimation. We address the reviewer comments below.
>
> ### Weaknesses
>
> > While at a high level the motivation of this work is good, the specifics of why this method enables better prediction and ultimately, better autonomous vehicle behavior, could be significantly strengthened. From the results, it seems like predictive performance is good for the baseline predictors (even marginally better than the proposed method) which would indicate that the baselines may be successful at prediction even on the presented OOD data (and therefore enabling downstream robot behavior to be safe /effective). This implies that there may be marginal benefits to doing OOD detection with the proposed method for the proposed scenarios. Intuitively, I’d imagine that if the predictor is presented with truly OOD data that is really difficult for it to generalize from the training data, then the prediction performance would significantly suffer.
>
> This is an excellent point! We believe there may be some confusion here. The performance reported in Table 1 is from the in-distribution test set, and not from the OOD data. The purpose of this experiment was to compare the trajectory prediction performance of uncertainty-aware approaches to the original CoverNet model, with the goal of maintaining the baseline’s performance. We have clarified this in the paper.
>
> To support the validity of our choice of OOD data split, we evaluated the CoverNet baseline on the OOD test set for the trajectory prediction metrics. The results are shown in the small table below on ID (OOD) data, which we can include in the appendix. There is a significant drop in CoverNet performance for both the input trajectory and map experiments when going from ID to OOD data. Thus, detecting these OOD examples would be important for safety critical applications.
>
> |Experiment | minADE$_{1}$      |  FDE
> |------------ | ----------- | ----------- |
> Input past trajectory    |  4.327 (7.130)    |  9.474 (13.632)
> Map | 4.732 (6.111) | 10.590 (13.464)
>
> > This suggests that perhaps this ID/OOD data split isn’t the best to showcase why measuring epistemic uncertainty is so necessary. For example, I’d posit that slow vs. fast variations in driving behavior are easy to generalize between and seen so frequently / in such large quantities in typical datasets, that SOTA predictors don’t struggle with speed of driving. However, OOD data like drunk drivers, U-Turns, construction zones, etc. intuitively seem like they would be very hard for SOTA predictors to deal with, thereby resulting in poor predictive performance and motivating the need for OOD detection. This paper would be strengthened by highlighting (1) ID/OOD data that SOTA predictors truly struggle with and (2) demonstrating how the proposed OOD method can enable, for example, more “safe” predictions by incorporating robot decision-making that takes safe recourse when presented with an OOD scenario.
>
> While we agree with the reviewer that OOD data like drunk or erratic driving would be relevant and interesting to consider, unfortunately the currently available open source trajectory prediction data does not contain significant examples of such rare OOD data. Hence, we thought about how we could use existing datasets (i.e., NuScenes) to showcase the ability of our proposed framework to distinguish OOD data. Since the uncertainty estimation we conduct in ISAP is task-aware, it was important that the OOD split affected trajectory prediction performance. Our OOD split ensures that trajectory prediction performance is affected when switching from ID to OOD data as shown in the table above.

---

> > ### Author Response · Authors · 2022-08-19
> > **Response to Reviewer WAfv (2/N)**
> >
> > > In general, it is challenging to understand the connection between the input x, the latent states z, and the generation of the dirichlet parameters α (which are the crux of the uncertainty quantification method). This makes it difficult to deeply understand the key idea of this work and its strengths/weaknesses.
> >
> > We are disappointed to hear that the reviewer found the relation between $x$, $z$, and $\alpha$ confusing. The ‘Input Representation’ and the $[v, a, h]$ state in Fig. 1 refer to $x$. The input contains a rendering of the map, the vehicle of interest (red) along with their past trajectory and current state, and the surrounding vehicles (yellow) along with their respective past trajectories. This is described in lines 111-119 in the paper. We have added notation from the paper into Fig. 1 (see attached image) as suggested by the reviewer.
> >
> > The input $x$ is encoded into the three latent variables: $z_{\text{agent}}$, $z_\text{map}$, and $z_\text{sc}$ as shown in Fig. 1 and described in lines 148-158. The interpretability for each of these latent variables is enforced using self-supervised decoders that attempt to reconstruct the semantic ideas from each latent variable.
> >
> > Normalizing flows are then learned over each of the $z_{\text{agent}}$, $z_\text{map}$, and $z_\text{sc}$ latent variables. The output of the normalizing flows are the densities $r(z^{(i)} \mid c; \phi)$, which are scaled to form the pseudo-counts $\beta_c^{(i)}$ in Eq. 2. Finally, the pseudo-counts $\beta^{(i)}$ are added to the prior $\beta^\text{prior}$ to form the output Dirichlet parameters $\alpha_{\text{agent}}$, $\alpha_{\text{map}}$, and $\alpha_{\text{sc}}$ for each of the three latent variables as indicated in lines 133-135 and lines 159-161.
> >
> > While the Dirichlet distribution provides an indication for the epistemic uncertainty of the network, the aleatoric distribution $\bar{p}^{(i)}$, parameterized by the normalized $\alpha$ parameters provides the categorical distribution over the discrete trajectory anchors. The predicted trajectory can be taken as, for example, the most likely trajectory anchor according to this categorical distribution, as in a classification task. We indicate that we consider trajectory prediction architectures with discrete anchors in lines 116-119. We discuss the output categorical distribution in lines 120-135 and describe our ISAP output in lines 159-161.
> >
> > > For example, it would be helpful to start with how PostNet takes as input x (and provide intuition for what this input is – e.g., this is a top-down RGB image of the scene) and this input is converted into a single latent state z via a neural network: z=fθ(x).
> >
> > For PostNet [34], the input x is arbitrary (e.g., MNIST or CIFAR-10 images or tabular data). In our paper, we are focusing on the top-down rendering representation and the state $[v, a, h]$ as the input $x$ for the trajectory prediction task as described in lines 111-119 and now also illustrated in Fig. 1.
> >
> > > Then, how does this latent state then get converted into trajectory predictions? The text never clearly states this.
> >
> > Please see above for where we summarize how the latent state gets converted to trajectory predictions and lines 130-135. However, we would be happy to add an additional sentence before line 162, to reiterate that the resulting $\alpha$ parameters get converted to a categorical distribution to output the most likely trajectory anchor according to Eq. 1.
> >
> > > Then, you can mention how to obtain uncertainty estimates: PostNet starts by learning a distribution over the latent state, conditioned on a discrete anchor class, c. Here it would be very helpful to provide intuition for why this anchor class is needed – for example, what does it represent in trajectory prediction?
> >
> > The anchor class can represent different predicted trajectory possibilities (e.g., going straight, turning right, stopping, etc.). These anchors can be achieved, for example, using k-means as in MultiPath [37] or by constructing a set of trajectories with a certain level of coverage as in CoverNet [38]. We describe this in lines 116-119.
> >
> > > Then, you can mention that the shape of this distribution is key to obtaining an uncertainty estimate. The epistemic uncertainty is modeled as a Dirichelt distribution with parameters alpha where α=βprior+βc. Here is where you leverage the distribution over the latent state: βc=Nc⋅r(z|c,ϕ). As the probability of a latent state given the anchor class goes to zero, then α approaches βprior. Here it would be helpful to provide intuition for why this is a desirable property.
> >
> > With lower confidence (smaller $\alpha$ parameters), we would hope that the learned distribution would go towards complete uncertainty corresponding to the uniform prior with parameters denoted as $\beta_\text{prior}$. We are happy to add a sentence after line 135 to highlight this point.

---

> > > ### Author Response · Authors · 2022-08-19
> > > **Response to Reviewer WAfv (3/N)**
> > >
> > > **Comment:**
> > >
> > > > Then, the ELBO loss can be clearly introduced as the method for how this network is trained. It would also be helpful to provide some intuition for this loss as well.
> > >
> > > We provide context for the ELBO loss in lines 479-485 in the Appendix.
> > >
> > > ### Issues
> > >
> > > > Figure 1: There are several aspects of this figure that could be improved to help the reader understand the method. First, the inputs [v,a,h] are never described in the figure nor in the text. It could also help to label the input representation with x, as is referred to in the text. Second, it would be helpful to explicitly highlight where the latent states zagent, zmap, and zsc are obtained in the pipeline, since these are key to the contribution. Additionally, the block called “Backbone” could be more descriptive. Finally, it could be clarified where the α Dirichlet parameters come from, since this is another aspect of the contribution. It is also unclear why there are only three predictions as output of the final Dir block? Are these just the three most likely trajectories, or does the method only allow for three predictions? Overall, making the text notation consistent and explicit in Figure 1 would help this figure clarify + enhance the technical content.
> > >
> > > Thank you for these suggestions on Fig. 1! We have incorporated them and as such improved the figure (**see in attachment**)!
> > >
> > > The input $x$ includes both the input top-down view representation and the current state $[v, a, h]$. The $[v, a, h]$ state, which contains the speed, acceleration, and heading change rate, and the complete input $x$ are described in lines 111-115. We have now also added an explicit $x$ label to Fig. 1 as suggested.
> > >
> > > We have added labels for $z_{\text{agent}}$, $z_{\text{map}}$, and $z_{\text{sc}}$, into Fig. 1. We left the ‘Backbone’ block as is since any image feature extractor could be used here. For example, we use ResNet-18 (see Appendix A, lines 465-472).
> > >
> > > We have also included $\alpha_{\text{agent}}$, $\alpha_{\text{map}}$, and $\alpha_{\text{sc}}$ labels in Fig. 1. We have now indicated in the caption that the three trajectories are exemplary most likely trajectories output by ISAP. We hope that these additions in combination with the caption description makes the connection clear.
> > >
> > > > In the summary of PostNet, it was also difficult to understand what the index (i) stands for. Please clarify this.
> > >
> > > The index (i) stands for the ith example in the dataset. We have added further clarification for this beyond line 128 in line 129.
> > >
> > > > In equation (4) why is it desirable to always equally weigh the three alpha parameters? Providing some intuition here would be helpful.
> > >
> > > It is not necessarily desirable or not to equally weigh the three alpha parameters. We chose a uniform prior here for simplicity. But if there is prior knowledge that, for example, the input past trajectory is most important to consider for the trajectory prediction task, the weights could be changed accordingly.
> > >
> > > > Please clarify what each aspect of agent state means: [v,a,h]. Also, is this just an instantaneous state or do you obtain the states over time as well?
> > >
> > > We define the state $[v,a,h]$ in lines 114-115. It corresponds to speed, acceleration, and heading change rate. This is the agent’s current state (line 113), so it is the most recently available instantaneous state.
> > >
> > > > Are these “types” of OOD data something that is typically a problem for predictors? It would be more powerful to first present the key “types” of OOD data that often mess up ensembles or other baselines. This would make the types of data used for testing much more convincing. Otherwise, right now it seems like somewhat of an ad-hoc choice to say that in-distribution data is “slow driving” while OOD data is “fast driving” when perhaps these types of variations in driving behavior are seen so frequently and in such large quantities in typical datasets, that SOTA predictors don’t struggle with them.
> > >
> > > Although we generally agree that more challenging OOD splits like erratic driving for trajectory prediction would be more interesting, however, we do not currently know of such data annotation that is available open source. Thus, we wanted to showcase the capabilities of our uncertainty-aware method on open source NuScenes data. We considered two data splits, one based on input trajectory speed and one based on map type (e.g., roundabout or not). The table included above confirms that these data splits are indeed challenging for the CoverNet baseline. These experiments also allow us to demonstrate the interpretability of the proposed ISAP approach.
> > >
> > > > Are the latent states z assumed to be discrete or continuous? It would be helpful to clarify this.
> > >
> > > The latent space z is continuous. We have added this information in line 137. The latent dimensionality of 4 is listed in lines 190-192.
> > >
> > > **Zip File:**
> > >
> > > /attachment/977cde45345e8ae0047b52c64b69ee3c322632da.zip

---

> > > > ### Author Response · Authors · 2022-08-19
> > > > **Response to Reviewer WAfv (4/N)**
> > > >
> > > > **Comment:**
> > > >
> > > > > ”Post-CoverNet is an ablation….without the interpretability element” does this just mean that there is no separation of the latent state into three components, and instead Post-CoverNet just produces a single latent state z? It would be helpful to clarify this.
> > > >
> > > > Yes, this is correct. We have added a clarifying sentence emphasizing the differences between ISAP and Post-CoverNet in line 207.
> > > >
> > > > > "The ground truth labels are 0 for incorrect predictions and 1 for correct predictions." It seems like the labels shouldn’t correspond to state prediction accuracy but whether or not a data point is ID or OOD. It would be helpful to clarify this in the text.
> > > >
> > > > Here we want to evaluate how well the confidence is calibrated. Specifically, we want the network to output high confidence for correctly predicted examples and low confidence for incorrect predictions. You are correct in that when we are evaluating OOD detection performance, the labels are 0 for OOD and 1 for ID. These metrics are described in lines 225-230. We have also added further wording with the above intuition into this section.
> > > >
> > > > > What does subscript c refer to? What are the possible anchor classes?
> > > >
> > > > The subscript $c$ refers to the discrete anchors described in lines 116-119. The subscript was defined in line 137, but we have moved this up to line 129, right below Eq. 1. Thank you for pointing this out!
> > > >
> > > > Since we are using CoverNet [38] within our ISAP architecture, the anchors are a trajectory set with a particular level of coverage for the trajectories in the training data. We inherit this trajectory set from the CoverNet model. We used 64 trajectory anchors for our experiments. This information is described in lines 186-190. For more information on how the trajectory anchor set is obtained, please see the CoverNet [38] paper.
> > > >
> > > > > In the Brier score definition, p¯ and d seem to be incompatible: p¯ is defined as a distribution while d  is a class label. Also, what kind of classes can d represent? Are classes the type of agents (e.g., car, bike, truck etc.) or something else?
> > > >
> > > > Thank you for the catch! This is a typo: $\bar{p}$ should be replaced with $\bar{\xi}$, which are the parameters for the categorical distribution (see Eq. 1). The labels $d$ are represented as one-hot encodings (line 224) allowing us to compute the distance between the two vectors.
> > > >
> > > > The classes here and throughout the paper refer to the discrete trajectory anchors (lines 116-119). They correspond to different possible predicted trajectories. Intuitively, there is a class for going straight, slowing down, accelerating, turning, etc.
> > > >
> > > > > Figure 2: It was difficult to compare the leftmost plot (whose y-axis is total data count) vs. the right three plots (whose y-axis is the α0,agent) since they have different y-axis, but look as though all four subplots should be compared equally. It would be helpful to explain how to compare / interpret these plots, or make the axes all have the same units.
> > > >
> > > > Although it is indeed not possible to compare the true data counts and the pseudocounts $\alpha$ directly in magnitude, the intention of this figure is to see similar trends in the true data count distribution and the learned pseudocounts. In particular, we see clear similarities between the trends for the true data counts and the $\alpha_{\text{agent}}$ pseudocounts with increasing past trajectory distance (higher input agent trajectory speed). For example, there are two peaks in certainty in both figures: at 0 m (stopped) and 5 m (about 18 km/h). Note that we can convert this distance on the x-axis to an estimated speed because all input past trajectories for the agent of interest are collected over 1 second (see lines 173-174). After the OOD cut-off at 10 m distance, we see a sharp drop-off in certainty for the data, quickly decaying to complete uncertainty for the OOD input. Although the distinction between ID and OOD is not as obvious for the map and social context latent variables, we still see a general trend of lower certainty as the trajectory distance increases. We discuss these plots in lines 252-260.
> > > >
> > > > > Figure 3: It’s difficult to interpret the second plots from the left (with α0,agent). Is this a spatial visualization (so the x and y axis represent the x and y of the world) where the red line is some sort of prediction? Or do the axes represent something different?
> > > >
> > > > Fig. 3 shows the input representation and the decoded latent variables, which represent the input past trajectory for the agent of interest, the map, and the social context. The second plot from the left shows the decoded input past trajectory for the agent of interest over 1 second. Yes, the trajectory is represented in x-y coordinates. This information is discussed in the caption and in lines 261-274. However, we have now included axis scales for these plots and added units in the caption (**see the attached figure**).
> > > >
> > > > **Zip File:**
> > > >
> > > > /attachment/e5d4c9eb43ef05113a043c96c14664e293180227.zip

---

> > > > > ### Author Response · Authors · 2022-08-19
> > > > > **Response to Reviewer WAfv (5/N)**
> > > > >
> > > > > > Since the trajectory prediction for the proposed method seems to be on-par with the SOTA methods, it seems to imply that all the predictors can do a decent job of predicting the OOD examples too. So, even though the proposed method can detect OOD examples, the ultimate prediction performance (which would then affect the autonomous vehicle decision-making) seems to not need any OOD detection to achieve good ADE/FDE scores. I think these results warrant (1) investigation into “harder” OOD datasets that clearly showcase what SOTA predictors would suffer in prediction performance and (2) how being able to do OOD detection can be integrated with robot decision-making to achieve safer behaviors than what is achieved without OOD detection.
> > > > >
> > > > > As we mentioned above, there seems to be some confusion here. In Table 1, we evaluate the performance of the different approaches on in distribution (ID) data. The goal of Table 1 was to ensure that the trajectory prediction performance is not significantly compromised with the addition of uncertainty estimation. We have provided additional trajectory prediction results on OOD data above for CoverNet and will include this table in the appendix. We agree this information is helpful to confirm that the trajectory prediction performance of CoverNet is worse on our proposed OOD data than on ID data, warranting effort into OOD detection.

---

### Official Review · Reviewer_xJrB · 2022-07-31

**Originality:** Good
**Technical Quality:** Good
**Clarity Of Presentation:** Very Good
**Impact:** 3

**Recommendation:**

Weak Accept: I recommend accepting the paper, but will not argue for my recommendation if the majority of other reviewers have a different opinion.

**Summary:**

This work proposed a self-aware framework for motion prediction with interpretability of the estimated uncertainty. The proposed method, Interpretable Self-Aware Prediction (ISAP), is able to evaluate the epistemic uncertainty by introducing an evidential deep learning method and then distributing the epistemic uncertainty among several interpretable semantic concepts.

The key contributions of this paper are:

1. They invest in using evidential deep learning methods for uncertainty estimation for the trajectory prediction task.

2. They add interpretability into epistemic uncertainty estimations by distributing them over low-dimensional interpretable latent spaces.

The author shows good uncertainty estimation results over AUROC and APR, meaning that it can accurately estimate the epistemic uncertainty of the trajectory prediction model.

**Issues:**

* Could you please give more details about the training time as you mentioned in 285 they can be brittle and slow to train.
* Results shown in the paper seem can not show the learned interpretability of social context because of sparse interaction in Nuscenes, which is usually hardest to handle and where the uncertainty estimation is more useful. Therefore, maybe you can try WOMD/INTERACTION dataset which has more meaningful interactions instead of setting different coefficients during training.
* For the OOD dataset split, why do you choose the l2 distance between the most recent and the earliest waypoints, what if it is a slow turn instead of going straight? And is the difference big enough to consider it as out-of-distribution data? The map-based setting seems more reasonable.
* Maybe you can add more uncertainty estimation and OOD detection baseline to compare the performance of this particular framework.



**Quality Of The Limitations Section:**

Limitations are not well addressed

**Reviewer Expertise:**

3: The reviewer is fairly confident that the evaluation is correct

**Robotics Focus:**

Highly relevant to robotics but no hardware experiments

**Strengths And Weaknesses:**

It is useful to introduce self-awareness to the motion prediction module to let the model tell what is known and what is unknown since over-confident prediction results might be harmful to the downstream modules. Besides, existing methods lack the interpretability of the estimated uncertainty, and it is helpful to introduce semantic concepts to make robust and safe trajectory predictions.

However, more baseline of uncertainty estimation methods need to be shown in the experiment, and the influence of the semantic concept of social context is not obvious because of the lack of interactions in the chosen dataset. More discussions of the limitation are needed.

**Summary Of Recommendation:**

The paper is well written and easy to read. Although the evidential deep learning framework is not original, readers can benefit from the self-aware motion prediction framework, which is rarely mentioned in this area. Furthermore, introducing interpretability is also useful to the downstream modules in the autonomy stack.

However, there is a need for a more developed discussion around the limits of the approach.

---

> ### Author Response · Authors · 2022-08-19
> **Response to Reviewer xJrB**
>
> We thank the reviewer for their thoughtful comments and suggestions. We were happy to see that the reviewer found our ISAP approach useful both in terms of uncertainty estimation for trajectory prediction and interpretability of said uncertainty, as well as our paper ‘well written and easy to read’. We address the reviewer comments below.
>
> > Could you please give more details about the training time as you mentioned in 285 they can be brittle and slow to train.
>
> For the input past trajectory experiment, the training time was about 37 hours on a single NVIDIA GeForce RTX 2080 Ti GPU including some extensive writing to TensorBoard during training, which significantly impacted the training time. For the map-based experiment, the training time was 18 hours on the same single GPU. More details on the experimental set-up and data set sizes are in Sec. 4 and the Appendix.
>
> > Results shown in the paper seem cannot show the learned interpretability of social context because of sparse interaction in Nuscenes, which is usually hardest to handle and where the uncertainty estimation is more useful. Therefore, maybe you can try WOMD/INTERACTION dataset which has more meaningful interactions instead of setting different coefficients during training.
>
> Although we generally agree that NuScenes has sparse highly interactive scenarios, the dataset does have general social context information (e.g., driving in traffic versus on an open road, turning cars, etc.). Thus, we are able to use this dataset, which is widely employed in trajectory prediction literature, to demonstrate the capabilities of our proposed ISAP approach. We discuss the social context implications for our qualitative examples in Sec. 5 and Appendix B.2.
>
> > For the OOD dataset split, why do you choose the l2 distance between the most recent and the earliest waypoints, what if it is a slow turn instead of going straight? And is the difference big enough to consider it as out-of-distribution data? The map-based setting seems more reasonable.
>
> Great questions! We gave significant thought to the choice of OOD split in the data. Since the uncertainty estimation we conduct in ISAP is task-aware, it was important that the OOD split affected trajectory prediction performance. The relative speed of the trajectory, regardless of a turn or straight driving maneuver, provided a significant difference in trajectory prediction for the CoverNet baseline between the two data categories as shown in the table included below on ID (OOD) data. Similarly, the map experiment provided a smaller, yet still significant performance difference as well.
>
> |Experiment | minADE$_{1}$      |  FDE
> |------------ | ----------- | ----------- |
> Input past trajectory    |  4.327 (7.130)    |  9.474 (13.632)
> Map | 4.732 (6.111) | 10.590 (13.464)
>
> Furthermore, for the input trajectory experiment, the $\ell_2$ distance is a continuous measure which allows us to see how the \alpha values evolve with the increase in $\ell_2$ in Fig. 2. ISAP’s ability to clearly distinguish between the two data categories (see quantitative results in Table 2) further supports this choice of data split.
>
> We will also move the map-based experiment into the main paper. With these two experiments, our goal was to demonstrate the usefulness of the interpretability element to our proposed ISAP approach.
>
> > Maybe you can add more uncertainty estimation and OOD detection baseline to compare the performance of this particular framework.
>
> We chose ensembles (of two sizes) as our baselines since ensembles are generally regarded as the state-of-the-art for OOD detection [26] and have previously been employed in robotics settings (e.g., [29])].

---

### Comment · Area_Chair_6Wat · 2022-08-19
**Meta Review of Paper54 by Area Chair 6Wat**

Thank you authors and reviewers, here is a short summary of some of the key strengths and weaknesses identified by the reviewers.
Authors & reviewers, please engage in a discussion regarding the issues raised by the individual reviews.

Strengths:
- the paper's research objectives in the area of epistemic uncertainty quantification for trajectory prediction were considered to be timely   and of important relevance
- the evaluation on the NuScenes dataset was appreciated
- experiments indicate competitive trajectory prediction with improved uncertainty estimation

Weaknesses:
- concerns regarding the potentially incremental nature of the approach were raised regarding similarities to PostNet
- elaborations on limitations of the approach were requested
- concern regarding the potentially small benefit of OOD detection in the considered dataset and the approach to reasoning about OOD in terms of speed were raised
- clarity of exposition (e.g. regarding PostNet and clarification of connection to Dirichlet parameters) could be improved further

---

> ### Author Response · Authors · 2022-08-20
> **Response to Area Chair 6Wat**
>
> Thank you for the helpful summary! We were happy to see that the reviewers found our Interpretable Self-Aware Prediction (ISAP) method to be interesting and useful, our evaluation thorough with ‘good baseline selection’ showing ‘superior uncertainty estimation performance’, and our paper ‘well written and easy to read’ with ‘clean exposition’. We have posted our initial responses to the reviewers and are looking forward to the discussion! We summarize our responses to the listed weaknesses below.
>
> > concerns regarding the potentially incremental nature of the approach were raised regarding similarities to PostNet
>
> Our paper's aim is to define a framework for epistemic uncertainty-aware trajectory prediction in the context of AVs. The goals for ISAP were to achieve one-shot epistemic uncertainty estimation working towards real-time capabilities and ensure that the resulting uncertainty is usable and interpretable by other autonomy stack components and engineers for further model development. These characteristics, to our knowledge, have not yet been achieved in epistemic uncertainty estimation for trajectory prediction, particularly as this is still an early research direction.
>
> Evidential deep learning methods, the class of approaches that PostNet belongs to, are able to estimate epistemic uncertainty in one-shot, and PostNet moreover does not require explicit OOD data during training. These attributes make PostNet a suitable backbone to build upon for our ISAP approach. PostNet [34] was originally only validated on small classification tasks (e.g., MNIST and CIFAR10), while trajectory prediction is a much more challenging, large-scale problem for which PostNet has not previously been explored. Our novelty lies in recognizing that to make the approach applicable and useful to the trajectory prediction task, we need to infuse interpretability into the estimated uncertainty, forming the ISAP framework. We empirically demonstrate that our proposed semantic breakdown of the uncertainty in the trajectory prediction task improves ISAP’s ability to identify OOD examples.
>
> > elaborations on limitations of the approach were requested
>
> We have added a discussion to the limitations of the intentional bias we impose on ISAP in terms of application within the AV domain as suggested by reviewer we25. Specifically, we design ISAP with trajectory prediction in the context of AVs in mind. If we were to consider trajectory prediction in an indoor environment with a mobile robot, a lot of the same semantic splits could be considered. However, for an application like assistive robotics, a different semantic breakdown might make more sense that would focus on task-level concepts (e.g., cooking, cleaning) and types of rooms (e.g., kitchen, living room, bathroom).
>
> > concern regarding the potentially small benefit of OOD detection in the considered dataset and the approach to reasoning about OOD in terms of speed were raised
>
> We gave significant thought to the choice of OOD split in the data. Since the uncertainty estimation we conduct in ISAP is task-aware, it was important that the OOD split affected trajectory prediction performance. The relative speed of the trajectory provided a significant difference in trajectory prediction for the CoverNet baseline between the two data categories as shown in the table included below on ID (OOD) data. We will include this table in the appendix.
>
> Furthermore, the $\ell_2$ past trajectory distance is a continuous measure which allows us to see how the \alpha values evolve with the increase in $\ell_2$ in Fig. 2. ISAP’s ability to clearly distinguish between the two data categories (see quantitative results in Table 2) further supports this choice of data split.
>
> In addition, we consider a map-based split in Appendix B, where we keep left-side driving in Singapore and avoid big streets and roundabouts for in distribution data. For OOD, we use data from Boston (right-side driving) and ensure ‘roundabout’ is in the scene description (lines 508-511). We are currently incorporating this experiment into the main paper.
>
> With these two experiments, our goal was to demonstrate the usefulness of the interpretability element to our proposed ISAP approach.
> While we could use corruption or adversarial attacks to achieve OOD examples, we were looking to validate our OOD approach on naturally occurring and more plausible OOD data.
>
> |Experiment | minADE$_{1}$      |  FDE
> |------------ | ----------- | ----------- |
> Input past trajectory    |  4.327 (7.130)    |  9.474 (13.632)
> Map | 4.732 (6.111) | 10.590 (13.464)
>
> > clarity of exposition (e.g. regarding PostNet and clarification of connection to Dirichlet parameters) could be improved further
>
> We have clarified the points of confusion indicated by reviewer WAfv in our response. We have also incorporated their suggestions into our process figure to further improve clarity for how our proposed ISAP framework computes the interpretable Dirichlet parameters.

---

### Author Response · Authors · 2022-08-24
**Updated Manuscript**

We have incorporated the suggestions provided by the reviewers into an updated version of our paper (see attached). Our changes are denoted in blue. The key changes are as follows:

* The map-based experiment has been moved into the main paper from the appendix.
* We have added validation (see Appendix C) that our chosen OOD splits are indeed difficult to generalize to for the CoverNet trajectory prediction baseline. Thus, it is important to detect OOD examples in these settings.
* Table 2 now includes both ID and OOD results for the uncertainty/calibration metrics.
* We have added the expected calibration error (ECE) metric to our results in Table 2.
* We have included entropy plots in Appendix D.
* Fig. 1 has been updated to include the notation used in the body of the paper.
* We have provided further clarifications throughout the paper as suggested by the reviewers to improve clarity of exposition.
* The limitations section has been expanded.
* We have fixed the four typos noted by the reviewers.

Please note that (1) the line numbers in our responses to reviewers refer to the originally submitted manuscript, not the updated version, and (2) during the rebuttal stage, the updated paper can be over 8 pages of main content (ours is currently 9 pages).

We would like to thank the reviewers and the area chair for their thoughtful comments, which have improved our paper! We are looking forward to further discussions.

---

### Meta-Review · Area_Chair_6Wat · 2022-09-05

**Recommendation:** Accept (Poster)
**Confidence:** 4

**Metareview:**

Concluding comments:

The authors present a clearly written framework for epistemic uncertainty estimation with significance in particular to safety critical applications of trajectory prediction methods in autonomous driving. The underlying problem of epistemic uncertainty estimation is clearly of relevance to the community and the presented approach appears to be quite well founded and evaluated on the NuScenes dataset.

The reviewers were uniform in their recommendation of weak accept x4. Following the initial reviews, the authors have further addressed the concerns from the previous meta review (see below) in their revision/rebuttal to a point where I would also like to recommend the work to be accepted.